# Enhanced SSVEP Bionic Spelling via xLSTM-Based Deep Learning with Spatial Attention and Filter Bank Techniques

**DOI:** 10.3390/biomimetics10080554

**Published:** 2025-08-21

**Authors:** Liuyuan Dong, Chengzhi Xu, Ruizhen Xie, Xuyang Wang, Wanli Yang, Yimeng Li

**Affiliations:** 1Hubei Provincial Key Laboratory of Green Intelligent Computing Power Network, School of Computer, Hubei University of Technology, Wuhan 430068, China; dly04100228@gmail.com (L.D.); xrz203359@outlook.com (R.X.); wxy326634@outlook.com (X.W.); 2School of Artificial Intelligence and Automation, Huazhong University of Science and Technology, Wuhan 430074, China; wly201008@outlook.com; 3School of Computer, Central China Normal University, Wuhan 430079, China; yimengl2035@gmail.com

**Keywords:** steady-state visual evoked potentials, brain–computer interface, xLSTM, attention mechanism, filter bank, multi-scale feature

## Abstract

Steady-State Visual Evoked Potentials (SSVEPs) have emerged as an efficient means of interaction in brain–computer interfaces (BCIs), achieving bioinspired efficient language output for individuals with aphasia. Addressing the underutilization of frequency information of SSVEPs and redundant computation by existing transformer-based deep learning methods, this paper analyzes signals from both the time and frequency domains, proposing a stacked encoder–decoder (SED) network architecture based on an xLSTM model and spatial attention mechanism, termed SED-xLSTM, which firstly applies xLSTM to the SSVEP speller field. This model takes the low-channel spectrogram as input and employs the filter bank technique to make full use of harmonic information. By leveraging a gating mechanism, SED-xLSTM effectively extracts and fuses high-dimensional spatial-channel semantic features from SSVEP signals. Experimental results on three public datasets demonstrate the superior performance of SED-xLSTM in terms of classification accuracy and information transfer rate, particularly outperforming existing methods under cross-validation across various temporal scales.

## 1. Introduction

In the fields of neuroscience and cognitive psychology, exploring how the brain processes and responds to external stimuli is crucial. With the continuous progress of bionics technology, we have been able to deeply understand the working principles of the brain, and applications such as spellers, robotic arms, and BCI medical recovery [1,2,3] have come into reality. As a special electroencephalogram (EEG) phenomenon, the SSVEP is a reaction of the visual cortex to periodic visual stimulation, which has the advantages of outstanding classification performance and strong anti-interference ability [4,5]. It employs specific frequencies of visual stimuli to trigger electrical changes in the brain at corresponding frequencies, which can be captured and analyzed by EEG devices, providing an efficient means of communication for non-invasive brain–computer interfaces. It can increase the SSVEP signal category by setting a variety of visual flicker stimulation frequencies, thereby building a rich instruction set to enable individuals with aphasia to communicate with the external world via biomimetic spelling paradigms [6].

Conventional SSVEP signal classification methods, such as Canonical Correlation Analysis (CCA) [7], are limited by noise sensitivity and linear signal characteristics, which makes them less effective at capturing phase information. To address these issues, several improved algorithms have been proposed. FB-CCA [8] firstly deployed the filter bank technique on CCA, combining fundamental and harmonic frequencies to enhance SSVEP detection. The more representative methods include TRCA and its variants [9,10,11], which improved the signal-to-noise ratio by maximizing task-based data reproduction and removing background EEG activity. Due to the issue of redundant spatial filters in TRCA, TDCA [12] employed a discriminative model that maximizes inter-class distinctions and reduces intra-class distinctions for feature extraction. In other ways, SS-CCA [13] and PRCA [14], respectively, improved performance by introducing temporal delay to signals and replicating periodic repeat components.

Recently, deep learning has driven significant innovation in BCIs by offering powerful nonlinear feature extraction, making it increasingly advantageous over conventional algorithms in solving complex classification tasks [15,16]. For time-domain data, EEGNet [17] and SSVEPNet [18] stand out as prominent models, both leveraging convolutional neural networks (CNNs) to recognize diverse stimulus signals. Similarly, the filter bank technique can be incorporated into deep learning methods as well [19,20]. However, time-domain signals are susceptible to noise and periodic signal observation challenges. For this, researchers have explored frequency features using Fast Fourier Transform (FFT) to analyze richer amplitude and phase information. The CCNN [21] proves the effectiveness of the combination of complex spectral features and convolutional neural networks. Subsequently, PLFA and MS1D_CNN [22,23], respectively, utilized spatial attention and Squeeze-and-Excitation [24] modules to enhance frequency features. To alleviate the problem of insufficient data, a 3DCNN [25] employed a deep transfer learning strategy on the frequency domain.

To reveal the complex topological relationships among EEG channels, graph neural networks have shown application prospects in the field of EEG due to their robust trustworthiness and ability to discover potential influences [26]. In the first application of EEG emotion recognition based on GCNNs [27], it adopted graph structure to model signals and capture the correlations between different regions of the brain. On this basis, a DGCNN [28] utilized the dynamic graph convolution method to learn the adjacency matrix between channels to evaluate the connection topology within the brain network. Specifically for SSVEP classification, the DDGCNN [29] introduced hierarchical dynamic graph learning and dense linking, and, respectively, employed dynamic convolutional kernels and graph dynamic channel fusion techniques to reconstruct the DGCNN and reduce computational complexity, further verifying the feature contributions of different brain regions to decoding.

Driven by the development of large language models, transformer-based models have been extensively applied in the fields of signal processing for their powerful sequence modeling capability and have achieved excellent results [30,31,32,33]. However, the self-attention mechanism of the Transformer [34] results in a reduction in computational efficiency and resource wastage due to redundant computation. The emergence of xLSTM [35], another novel large language model architecture, not only overcomes the limitations of traditional LSTM [36] models in handling long sequences and concurrent computations but offers different insights for the field of signal processing.

In this paper, a neural network, SED-xLSTM, is proposed for speller stimulus classification of SSVEPs, integrating xLSTM and spatial attention [37] with Transformer patterns to construct encoders and decoders, which are widely adopted in the realm of image segmentation. The approach initially maps frequency-domain information into high-dimensional semantic representations containing spatial-frequency and correlation features via two distinct encoders. Subsequently, the xLSTM-based decoder reconstructs the combined features and removes redundancy to ultimately accomplish the classification. Additionally, the filter bank technique is deployed before entering the SED-xLSTM to fully utilize harmonic frequencies for further improving performance. Furthermore, we compare SED-xLSTM with other baseline methods and replace xLSTM with another mainstream sequence model as well. The experimental results illustrate that SED-xLSTM exhibits distinct advantages in average accuracy and average information transfer rate (ITR) under cross-validation across four different temporal scales.

## 2. Datasets and Processing

Datasets 1 and 2 are Benchmark and BETA versions of the public SSVEP dataset provided by Tsinghua University [38]. The datasets utilize the mode of a virtual keyboard, covering 40 types of frequencies from 8.0 Hz to 15.8 Hz with a frequency interval of 0.2 Hz, which is shown in Figure 1a. In addition, four different phases (0, 0.5π, 1π, 1.5π) are employed to modulate these frequencies. The datasets provide references for developers to construct a decoding system that can rapidly and accurately map the collected SSVEP signals to the corresponding keyboard keys. Dataset 1 is collected from 35 participants with six test blocks for each participant, and Dataset 2 contains 70 participants with four test blocks in the experiment. Each subject is required to gaze at the designated stimulus target to generate a detectable SSVEP response, which is then collected by the recording device. The sampling rate is 250 Hz and the EEG signals are collected using a 64-channel recorder, which consists of 0.5 s before, during, and after the stimulation. In Dataset 1, the stimulus duration is 5 s, whereas in Dataset 2, it is 2 s for the first 15 participants and 3 s for the others.

Dataset 3 is from the publicly available database UCSD in San Diego [39]. The dataset is derived from 10 healthy subjects, containing 12 frequency stimuli ranging from 9.25 Hz to 14.75 Hz with an interval of 0.50 Hz and four phases of 0, 0.5π, 1π, and 1.5π and is shown in Figure 1b. Each participant performs experiments with 12 different stimulus frequencies and a 256 Hz sampling rate. Every experiment consists of 15 separate tests, and each test contains 1114 sampling points.

For Datasets 1 and 2, the experiment selects all subjects in Dataset 1 and the last 55 subjects in Dataset 2 [22] and chooses 16 types of stimulus signal data out of 40 stimuli from all test blocks of each subject. In addition to the eight typical stimulus frequencies [22,23], we also select eight extra stimulus types with larger intervals from these typical stimuli as much as possible to enhance the discrimination of the stimulation samples in the process of adding classification tasks. The frequency and phase of the 16 stimuli range from 8.2 Hz to 15.2 Hz and from 8.6 Hz to 15.6Hz, with an interval of 1 Hz. In addition, we choose a total of nine electrodes (O1, O2, PO3, Oz, PO4, PO5, PO6, POZ and PZ) [40] and cut out 0.5 s before and after the stimulation in order to keep the continuity of the stimulus signal. Hence, the retained lengths of the signal in two datasets were 5 s and 3 s. For Dataset 3, since the stimulation starts from the 39th sampling point (after 0.15 s), our experiment selects all 12 stimulus frequencies and 4s stimulus signals corresponding to 0.15 s–4.15 s from eight electrodes (O1, O2, PO3, Oz, PO4, PO7, PO8, POZ) [21]. The signal clipping process corresponds to Figure 2b.

After the above operation, the EEG signals of the three datasets are filtered using a bandpass FIR filter with a range from 6 to 50 Hz, implemented using EEGLAB toolbox in Matlab. Since every subject has different response onset time and duration for the same visual stimulus frequency, Figure 2c,d adopt four different time window lengths of 0.5 s, 1 s, 1.5 s, and 2 s. Signals of each length are segmented according to a sliding window of 0.5 s until the end position of the last segment is the same as that of the original signal. The length of the overlapping part is the difference between the four time windows and 0.5 s. Segmenting the data with different time lengths not only enhances system stability but allows for the expansion of smaller datasets. The time domain feature dimension obtained by segmentation is (C,Sp), where *C* is the number of channels and Sp is the sample points.

Subsequently, we carry out an FFT transformation of 512 frequency points for each channel of each sample. When the input signal is real-valued, the FFT result exhibits conjugate symmetry, which implies that the second half of the spectrum is a mirror image of the first half. Therefore, we can obtain the real and imaginary parts of the spectrum by employing only half of the FFT result, specifically 256 points. The conversion expression is defined as(1)$$\begin{array}{c} FFT(x)=Re[FFT(x)]+iIm[FFT(x)] \end{array}$$
where *x* represents time domain information, *i* represents imaginary unit, Re and Im represent the real and imaginary parts of the complex spectrum, respectively. The amplitude information is obtained by calculating the modulus of the complex number through real and imaginary parts, while the phase is determined by the arctangent of the ratio of the imaginary part to the real part. The information of the two vectors is spliced into a 3D frequency domain feature matrix according to the dimensions of channels and frequency points. Through this, the dimension of the input data is (2,C,F), where the value of *F* is 256 and 2 represents the two dimensions of the real and the imaginary part, as illustrated in Figure 2e,f.

## 3. Preliminaries

xLSTM [35] is an extension of LSTM (Long Short-Term Memory), which introduces a new gating mechanism and memory structure to enhance the performance of LSTM in natural language processing. In the biological nervous system, neurons control the flow of information by regulating the strength of synapses and the efficiency of signal transmission. xLSTM inherits the brain-like memory pattern of LSTM, simulating the selective memory and forgetting of the brain through gate structure and cell state, thereby realizing flexible management of long and short-term memory. Owing to the favorable scalability, xLSTM has been gradually applied in the fields of image [41,42,43] and signal processing [44,45].

xLSTM is composed of the stacking of sLSTM [35] and mLSTM [35], which are variants of LSTM. sLSTM adds a scalar update mechanism on the basis of LSTM, employing exponential gating and normalization technique to optimize the accuracy and stability of the model. Meanwhile, mLSTM extends the vector operations from a scalar c∈R to a matrix $C\in R^{\;d\times d}$, enhancing the memory ability of the model and enabling it to perform parallel computing on data. Since mLSTM has no interactions between hidden states across successive timesteps, it can be fully parallelized on modern hardware to achieve fast computation. In this paper, the mLSTM is utilized as the basic module, which is defined as$$\begin{array}{cc} C_{t}=f_{t}\;C_{t-1}+i_{t}\;v_{t}\;k_{t}^{⊤} & \mathrm{cell}\;\mathrm{state}\;(2)\\ n_{t}=f_{t}\;n_{t-1}+i_{t}\;k_{t} & \mathrm{normalizer}\;\mathrm{state}\;(3)\\ h_{t}=o_{t}⊙{\tilde{h}}_{t},\;\;\;\;{\tilde{h}}_{t}=C_{t}\;q_{t}\;/\;\max\left\{\left|n_{t}^{⊤}\;q_{t}\right|,1\right\} & \mathrm{hidden}\;\mathrm{state}\;(4)\\ q_{t}=W_{q}\;x_{t}+b_{q} & \mathrm{query}\;\mathrm{input}\;(5)\\ k_{t}=\frac{1}{\sqrt{d}}W_{k}\;x_{t}+b_{k} & \mathrm{key}\;\mathrm{input}\;(6)\\ v_{t}=W_{v}\;x_{t}+b_{v} & \mathrm{value}\;\mathrm{input}\;(7)\\ i_{t}=\exp\left({\tilde{i}}_{t}\right),\;\;\;\;\;{\tilde{i}}_{t}=w_{i}^{⊤}\;x_{t}+b_{i} & \mathrm{input}\;\mathrm{gate}\;(8)\\ f_{t}=\sigma \left({\tilde{f}}_{t}\right)\mathrm{OR}\exp\left({\tilde{f}}_{t}\right),\;{\tilde{f}}_{t}=w_{f}^{⊤}\;x_{t}+b_{f} & \mathrm{forget}\;\mathrm{input}\;(9)\\ o_{t}=\sigma \left({\tilde{o}}_{t}\right),\;\;\;{\tilde{o}}_{t}=W_{o}\;x_{t}+b_{o} & \mathrm{output}\;\mathrm{input}\;(10) \end{array}$$
where $C_{t}\in R^{\;d\times d}$ denotes the cell state, $n_{t}\in R^{\;d}$ is the normalizer state, and $h_{t}\in R^{\;d}$ represents the hidden state. Additionally, mLSTM stores a pair of vectors, the key $k_{t}\in R^{\;d}$ and the value $v_{t}\in R^{\;d}$, to achieve higher separability according to the ccovariance update rule [46]. The forget gate controls the attenuation rate, the input gate regulates the learning rate, while the output gate scales the retrieval vector. Moreover, mLSTM invokes an additional state $m_{t}$ to stabilize gates, avoiding overflow caused by exponential activation functions:$$\begin{array}{cc} m_{t}=\max\left(\log\left(f_{t}\right)+m_{t-1},\log\left(i_{t}\right)\right) & \mathrm{stabilizer}\;\mathrm{state}\;(11)\\ i_{t}^{\prime }=\exp\left(\log\left(i_{t}\right)-m_{t}\right)=\exp\left({\tilde{i}}_{t}-m_{t}\right) & \mathrm{stabil}.\;\mathrm{input}\;\mathrm{state}\;(12)\\ f_{t}^{\prime }=\exp\left(\log\left(f_{t}\right)+m_{t-1}-m_{t}\right) & \mathrm{stabil}.\;\mathrm{forget}\;\mathrm{state}\;(13) \end{array}$$

During the process of state update, xLSTM retains and extends the dual states and multiple-gate structure of LSTM, providing a stable structural foundation for long-term memory retention and fine-grained control of reading and writing. In contrast, the GRU simplifies the gate structure and compresses semantic information into a single vector, sacrificing some expressive capacity for greater conciseness and efficiency. Owing to the presence of multi-scale components in EEG signals, such as transient events and artifacts, it is easier for GRU to confuse long-term features when there is a strong noise. For the Transformer model, despite its proficiency in capturing global dependencies and the capability for parallel computation, it is prone to overlook the local structures of EEG frequency-domain features and lacks sensitivity to fine-grained information.

## 4. SED-xLSTM

### 4.1. Overview

The SED-xLSTM aims to classify SSVEP speller instructions based on the extended LSTM, and it is roughly composed of five main components: the spatial attention encoder, the xLSTM-based encoder EM-block and decoder DM-block, the feature recalibration, and the output layer. The spatial attention mechanism is derived from the CBAM module [37], which is commonly used in image processing to enhance the features of key regions. In this model, the encoders and the decoder all apply the stacked structure according to Transformer architecture to learn the spectrogram semantic information of the signal in a hierarchical manner, enabling the model to progressively deepen the understanding of the input content. Considering that the feature fusion process leads to the generation of redundancy, we optimize and calibrate the fused features adopting the feature recalibration technique [24] to adaptively adjust the weights before they enter the decoder. Furthermore, SED-xLSTM is trained and calibrated in a data-driven manner, and the recognition of speller commands is conducted in an offline experiment.

From the view of EEG electrode channels, due to the spatial correlation of SSVEP signals among various channels, neuronal activities in certain regions of the scalp may synchronously respond during the visual stimulus evoking process. Therefore, the model integrates spectral information from multiple channels into patch blocks within both branches for holistic processing, thereby capturing the synergy among local channels. On the other hand, the patch-based processing not only preserves the discrete attribute of the data but prevents the model from overly relying on specific channels. Consequently, the xLSTM is leveraged to learn the cooperative dependencies of these multi-channel features and contains the harmonic information as well.

### 4.2. Detailed Structure

The pipeline and parameters of the model are illustrated in Figure 3. As mentioned in Section 3, the original input dimension of the model is $x\in R^{\;2\times C\times F}$. Before entering the EM-block, we utilize a convolutional neural network to divide the spectral features into several fixed-size patches as ViTs [47], which are flattened and mapped to the $hidden\in R^{\;N\times D}$. *N* represents the number of patches, while dm corresponds to the embedding dimension that is consistent with *F*. Subsequently, we incorporate trainable positional encoding into each patch to learn the sequential dependencies and heighten the sensitivity of the model to positional information. As for the EM-block, it consists of two mLSTM layers and multiple perceptron layers in series, with skip connections implemented. In the mLSTM layers, the number and size of the parallel heads are set to 4 and 128. The height and width size of each patch are 3 (UCSD: 2) and 64.

The FFT can reveal the frequency components and periodic patterns in data that are often not apparent and difficult to capture directly from the time-domain data. Frequency-domain features are more discriminative, enabling the spatial attention mechanism to more accurately focus on key regions and patterns, thereby achieving more effective feature enhancement. Since the response frequency of the brain to stimuli is particularly prominent on the EEG channel, the data is propagated in the form of 3D features in the spatial attention encoder (SAE) to maintain spatial coherence. This module compresses the input content along the channel dimension via max pooling and average pooling, and subsequently maps the compression results to a spatial weight matrix through a convolutional network and the Tanh activation function. The matrix is element-wise multiplied with the input content to obtain the key feature information, which is fed into the multilayer perceptron with skip connections implemented as well. After the feature enhancement by this encoder, the data is ultimately mapped to $hidden\in R^{\;N\times D}$ by an additional patch embedding layer for feature fusion with the output of the EM-block.

The feature recalibration module optimizes the fusion process by introducing a dynamic weighting mechanism that allocates weights based on the importance and relevance of features. The fused features are squeezed via max pooling and average pooling along the patch dimension and then pass through two fully connected layers and GELU activation for excitation. The patch weights obtained from the excitation operation are applied to the original fused feature map, thereby suppressing unimportant and redundant information.

Within the DM-block, a gating mechanism-based network structure is implemented for xLSTM to control the storage and update of effective information, and two convolutional modules are integrated to enhance non-linear expressiveness. The kernel size and stride of the convolutional layers are respectively set to 3 and 1, and the parameters of xLSTM are identical to those in the EM-block. In addition to the primary modules, the model extensively incorporates layer normalization and dropout techniques to stabilize training and mitigate overfitting, and the final instruction classification task is accomplished by the output layer.

### 4.3. Model with Filter Bank

Apart from the fundamental frequency, SSVEP data contains multiple harmonic components that can serve as classification features as well. Based on this theory, the filter bank technique is leveraged on this model to further improve the classification performance, which is shown in Figure 4. To make the frequency index on the spectrogram express the feature information as richly as possible, we first filter the signal into *n* bands through bandpass filters and subsequently perform FFT on the *n* groups of signals to obtain the frequency-domain features. Compared with the method of obtaining sub-frequencies by zeroing out the unnecessary frequencies, this approach makes the interval frequencies of the adjacent two indexes on the spectrogram more refined. The amplitude and phase information are then concatenated along the channel dimension and a trainable convolutional layer is implemented to weight and reshape the spectrograms, resulting in an input shape of (n,N,D). In this experiment, the number of filter banks is set at 3, consistent with other research [25,40], to maintain a balance between model complexity and performance. For the filters, the lower cutoff frequencies are all set at 6 Hz, while the upper cutoff frequencies are 50 Hz, 64 Hz, and 80 Hz, respectively, covering the frequency range of 3 to 5 times the fundamental stimulation frequency. To mitigate the overfitting impact caused by the noise in the input features based on the filter bank, we adjust the double mLSTM layer to a single layer and increase the dropout rate to 0.4 in the decoder of SED-xLSTM.

### 4.4. Training Settings

The SED-xLSTM network is trained by default deploying the Adam optimizer and the cross-entropy loss function with a batch size of 64. The initial learning rate is set to 0.001. Additionally, the learning rate attenuation mechanism and early stopping mechanism are implemented during the training process. The learning rate will be halved if the validation loss does not decrease for 5 consecutive epochs, and training will be terminated if the loss does not decrease for more than 25 consecutive epochs. Additionally, the network, employing one Nvidia RTX 3060 GPU and two A10 GPU graphics cards to accelerate computation.

## 5. Performance Analysis

### 5.1. Baseline Methods

In this paper, we conduct a comparative evaluation of our approach against six other baseline methods for EEG signal classification: TRCA [9], CCNN [21], EEGformer [32], PLFA [22], EEGNet [17], and SSVEPNet [18]. TRCA is a traditional knowledge-driven method, while the others are data-driven methods based on deep learning.

TRCA enhances the signal-to-noise ratio of task-related EEG components significantly by maximizing the reproducibility of time-locked activity across task trials to identify the optimal weighting coefficients.

CCNN includes a convolution layer and a fully connected layer and takes the complex spectrum of the signal as input data, which confirms the potential advantage of spectrum representation in SSVEP decode tasks.

EEGformer employs a one-dimensional convolutional neural network to automatically extract features from the EEG channels and integrates the Transformer to successively learn the temporal, regional, and synchronous characteristics of the EEG signals.

PLFA introduces the spatial attention mechanism to enhance the discriminative frequency information.

EEGNet integrates depthwise separable convolution and spatial pooling techniques to achieve automatic feature extraction of EEG signals.

SSVEPNet adopts a CNN-BiLSTM network architecture, combining spectrum normalization and label smoothing techniques to suppress overfitting.

### 5.2. Comparison Experiment

The performance of the methods is measured using accuracy and information transfer rate (ITR) as metrics under the identical data preprocessing conditions. Each classifier is independently trained for a 16-class classification task on the first two datasets and a 12-class classification task on the last dataset, with five-fold cross-validation across various data lengths. The formula of ITR is defined as(14)$$\begin{array}{c} ITR=\frac{60}{T}\times \left[\log_{2}N+\log_{2}P+\left(1-P\right)\times \log_{2}\frac{1-P}{N-1}\right] \end{array}$$
where *N* denotes the number of classification targets, *P* is the accuracy rate, and *T* denotes to the selection time of a single target, expressed in bits/min.

Table 1, Table 2 and Table 3 display the average accuracy of the various methods across the three datasets at different time window lengths. Due to the increase in time duration, the periodic characteristics contained in the signals are more obvious. As the time window length is set to 2 s, the proposed method achieves the highest accuracy performance on all datasets, with values of 96.84±0.50%, 90.82±0.77%, and 91.50±1.03%, respectively. Compared with the second-place SSVEPNet on Datasets 1 and 3, the average accuracy is higher by 3.30% and 1.43%, while it outperforms the second-place EEGformer by 1.77% on Dataset 2. After preprocessing with the filter bank (FB), the performance of the model is 97.62±0.56%, 92.19±0.61%, and 93.11±0.92%, respectively. In all experimental scenarios, the proposed method attains the optimal classification accuracy and reveals remarkable distinctions between SED-xLSTM and the baseline methods and between FBSED-xLSTM and the other seven methods via paired sample *t*-tests with a degree of freedom of 3 (*: *p* < 0.05, **: *p* < 0.01, ***: *p* < 0.001).

Figure 5 illustrates the ITR metric for various methods. SED-xLSTM achieves optimal performance when the time length is set to 1s, with values of 149.56±2.69 bits/min and 115.47±4.89 bits/min on Datasets 1 and 3, respectively. Similarly, it exceeds the second-place SSVEPNet by 5.54 bits/min and 7.14 bits/min. On the other dataset, the model performs comparable performance in the cases of 1 s and 1.5 s, with values of 110.08±3.25 bits/min and 111.46±2.15 bits/min. The second-place EEGformer is 9.06 bits/min and 5.36 bits/min lower, respectively. Additionally, this trend is observed in FBSED-xLSTM as well. Its ITR can reach a maximum of 157.47±2.51 bits/min and 122.46±4.55 bits/min on Datasets 1 and 3, while the values for 1 s and 1.5 s on Dataset 2 are 114.68±2.52 bits/min and 115.92±1.83 bits/min, respectively.

Among the existing sequence models, LSTM [36], GRUs [48], and Transformers [34] are widely recognized as methods with excellent sequence modeling capabilities. Nonetheless, due to the difficulty of parallelizing computations and the limitation in capability to handle long sequences in LSTM and GRUs, as well as the extensive computations for Transformers to gradually learn local information, we take advantage of xLSTM to circumvent the aforementioned limitations of conventional models in processing sequential data. Figure 6 exhibits a comparative evaluation of xLSTM against other sequence modules under the condition of identical numbers of layers or heads across various datasets. The results reveal that xLSTM maintains optimal performance in the majority of cases. It is worth noting that the Transformer module performs with no remarkable distinction compared to xLSTM on Dataset 1, achieving accuracy (ITR) values of 45.06±0.95% (103.27±4.16 bits/min), 79.54±0.79% (148.17±2.80 bits/min), 94.25±0.61% (137.69±1.91 bits/min), and 96.68±0.47% (109.81±1.24 bits/min) across the four data lengths. The GRU also shows no significant distinction to xLSTM at the data length of 2 s on Datasets 2 and 3, with accuracy (ITR) values of 90.27±0.69% (94.78±1.46 bits/min) and 90.92±1.18% (84.95±2.44 bits/min), respectively.

Furthermore, the model size and the computational memory usage are employed as two extra metrics for comparing the distinctions between xLSTM and the Transformer module, which is illustrated in Figure 7. Since the fixed conversion points of FFT, the two metrics remain consistent at various data lengths of the same dataset. In Datasets 1 and 2, the number of parameters and memory usage for a batch of xLSTM module are 36.1 M and 427.04 MB due to the identical sample size. For the Transformer module, the values are 21.8 M and 366.47 MB, which are 14.3 M and 60.57 MB lower than xLSTM, respectively. In Dataset 3, the model size and memory usage of xLSTM are 37.0 M and 437.97 MB, while the results for the Transformer are 22.7 M and 380.41 MB, which are 14.3 M and 57.56 MB lower, respectively. In the multi-head mechanism of mLSTM, each head independently performs complex matrix operations, and the embedding dimension of the model is mapped to high dimension through projection blocks for forward propagation. In contrast, the projection dimension in the Transformer is relatively lower, resulting in a smaller computational capacity.

To further validate the stability of the model on untrained subjects and comprehensively consider two performance metrics, accuracy and ITR, cross-subject generalization experiments are implemented in this paper at time lengths of 1s and 1.5s. For each dataset, the subjects are evenly divided into five groups. The data of each group is successively utilized for validation, while the remaining parts serve as the training set. Table 4 displays the generalization results of FBSED-xLSTM. In the case of 1s, the average accuracy and ITR of the five subsets on each dataset are 70.99%, 60.12%, 70.51% and 119.86 bits/min, 88.29 bits/min, 101.40 bits/min, respectively. As for 1.5 s, the values are 85.78%, 76.50%, 80.12% and 114.18 bits/min, 91.81 bits/min, 87.11 bits/min, respectively. In a prior training strategy, the data distribution of the training and validation sets is more consistent. However, the two datasets come from various subjects in cross-subject training. The model overly learns the signal features of the training subjects, and the substantial individual differences diminish the generalization capacity of the model in new subject data.

### 5.3. Ablation Study

To verify the validity of the SAE- and xLSTM-based modules proposed in this paper, we designed and conducted a series of ablation studies in which we progressively removed certain components from the model to observe their impact on model performance at four time lengths, as shown in Figure 8. In experiments involving the removal of the EM-block and DM-block, the accuracy and ITR metrics demonstrate a marked decline in comparison to the intact SED-xLSTM model, with the magnitude of reduction being roughly equivalent. This outcome indicates that the EM-block and DM-block have comparable impacts on the performance of the model. Furthermore, in the absence of the SAE module, the performance of the method declines the most across all datasets, which manifests the effectiveness of feature fusion from two encoders. This ablation study emphasizes the necessity of integrating the three sub-networks simultaneously and underscores the significance of combining their functionalities to optimize overall performance.

Moreover, the class activation mapping is carried out in this experiment to explore how the spatial attention mechanism and filter bank enhance the discernment of the model. Figure 9 exhibits the class activation results of the amplitude characteristic information of the fixed sample on Dataset 3 at a stimulation frequency of 9.25 Hz. When the number of spatial attention encoders is 0, the model mainly focuses on the low-frequency region, and the harmonic information at high frequencies is not fully utilized. Additionally, similar frequencies around this stimulus affect the judgment of the model as well. As the number of encoder layers increases, the harmonic information in the high-frequency region is gradually utilized, and the model reduces its focus on similar frequencies. For the filter bank, the input of the model is the spectral features of three specific filtering ranges. The harmonic features are amplified and weighted by class activation to enhance the discrimination ability.

## 6. Discussion

The method proposed in this study aims to tackle the redundant computation in Transformer-based approaches by introducing a gating mechanism, thereby offering a new alternative for signal decoding. The SSVEP spectrogram is essentially channel-oriented sequence data. Similar to text data, it also has rich contextual dependencies internally. By virtue of the unique structure, xLSTM exhibits high sensitivity to noise within the signal, which enables it to achieve higher-precision classification. Meanwhile, since the spectrogram features lack the distinct standard characteristics like texture and color that are commonly found in natural images, we incorporated network integration into our chosen strategy to enhance the overall feature representation.

Despite the relatively superior performance of SED-xLSTM, it is subject to certain limitations. Firstly, the SSVEP is characterized by a continuous and periodic response to visual stimuli of specific frequencies. When the time length of the sample is relatively short, the number of signal cycles collected is limited. This leads to a decrease in frequency resolution and signal-to-noise ratio during spectral analysis, making it difficult to fully capture stable evoked frequency components. Additionally, the sample is more susceptible to interference, which can mask the true characteristics of the SSVEP. Therefore, the shorter time window restricts the full manifestation of the significant frequency characteristics of the SSVEP signal, especially at the time length of 0.5 s. In the second place, the 16 stimuli decoded instead of 40 stimuli conducted on the first two datasets in this paper affects the the generalizability of the model as well. Looking ahead, we will concentrate on instruction recognition within short time windows to enhance the real-time response rate of BCI systems and achieve a broader classification of stimulus frequencies.

## 7. Conclusions

This paper addresses the issues of the insufficient extraction of SSVEP frequency domain information by existing deep learning models and the redundant computations generated by Transformer models, proposing the SED-xLSTM model method, which is based on a novel gating mechanism, xLSTM, to enhance the classification performance of speller instructions on few channels. The model employs a stacked encoder–decoder architecture for instruction classification and incorporates spatial attention mechanisms for feature enhancement. To fully exploit harmonic information, we further enhance the performance of the model by utilizing filter bank techniques in the preprocessing stage. In this paper, we pioneer the application of xLSTM to the decoding of SSVEP-based spellers and introduce the design principles of the model and the signal preprocessing procedures in detail, bridging the gap between natural language and SSVEP signals. Comparative experiments across three public datasets with four different time lengths demonstrate that the SED-xLSTM model exhibits remarkable distinctions compared to other baseline methods. Moreover, the xLSTM module has been proven to possess considerable competitiveness as well when benchmarked against existing mainstream sequential model architectures in this experiment.

## Figures and Tables

**Figure 1 biomimetics-10-00554-f001:**
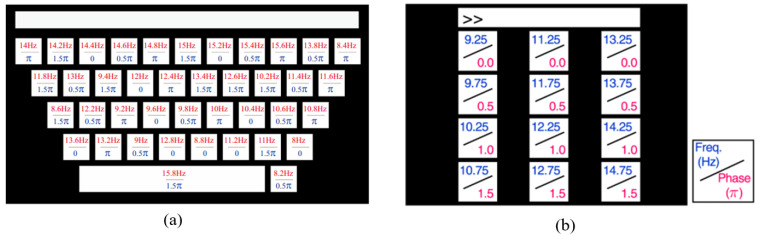
(**a**) The stimulus paradigm of dataset 1 and 2; (**b**) The stimulus paradigm of dataset 3. The numbers on the keyboard correspond to the flickering frequency of the display light source during the evoked experiment.

**Figure 2 biomimetics-10-00554-f002:**
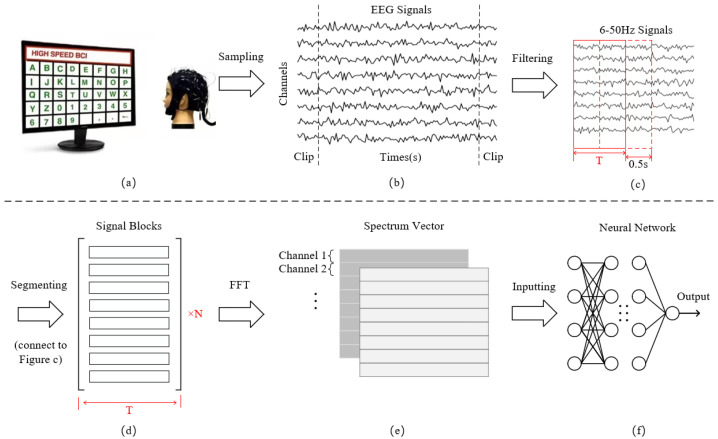
(**a**) Sample collection; (**b**) Signal filtering and extraction of stimulation periods; (**c**,**d**) Signal segmentation; (**e**) Fast Fourier Transform; (**f**) Decoding classification.

**Figure 3 biomimetics-10-00554-f003:**
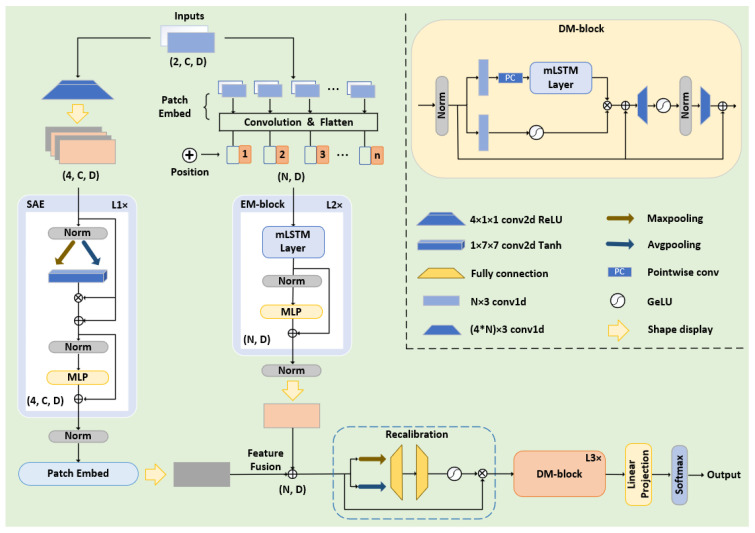
Detailed parameters and architecture of SED-xLSTM.

**Figure 4 biomimetics-10-00554-f004:**
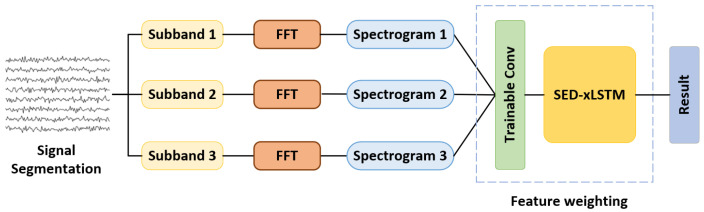
The filter branches obtained by the filter bank are weighted by a trainable convolutional layer to express harmonic information.

**Figure 5 biomimetics-10-00554-f005:**
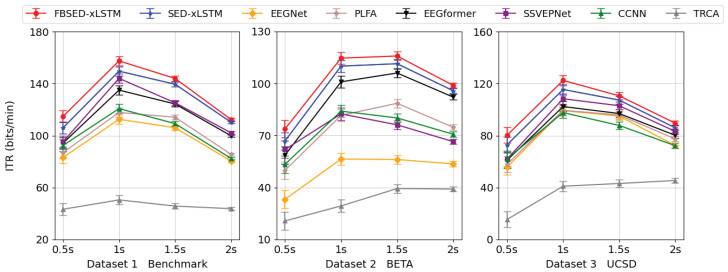
Average ITR of all methods at four time lengths for three datasets (bits/min).

**Figure 6 biomimetics-10-00554-f006:**
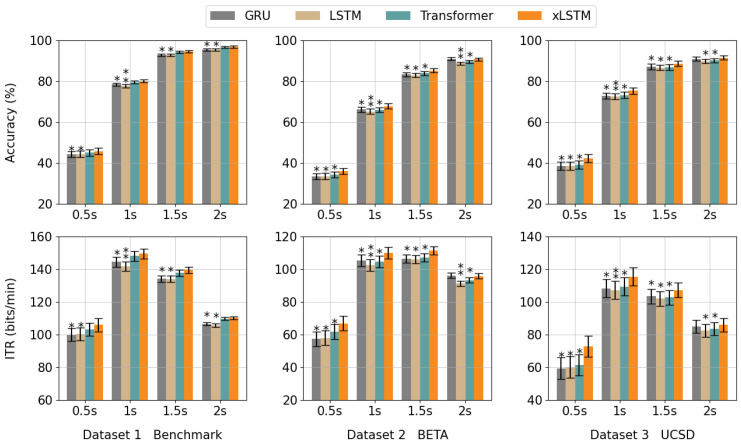
The first row represents the average accuracy of various sequence modules on each dataset, while the second row corresponds to the average ITR. The paired sample *t*-test with a degree of freedom of 3 is conducted (*: *p* < 0.05, **: *p* < 0.01).

**Figure 7 biomimetics-10-00554-f007:**
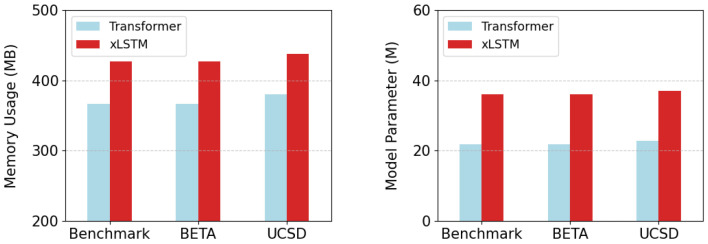
The comparison of the two modules in terms of memory usage and model parameters.

**Figure 8 biomimetics-10-00554-f008:**
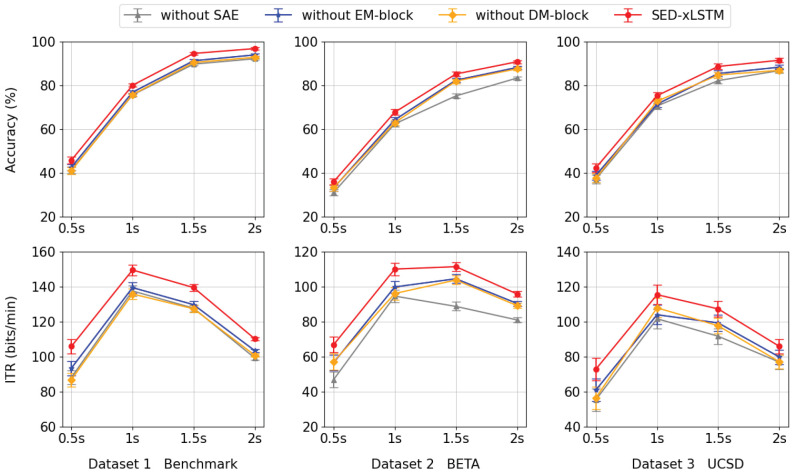
From left to right, the first row presents the accuracy results for each component across various datasets, while the second row shows the corresponding ITR values in the same sequence.

**Figure 9 biomimetics-10-00554-f009:**
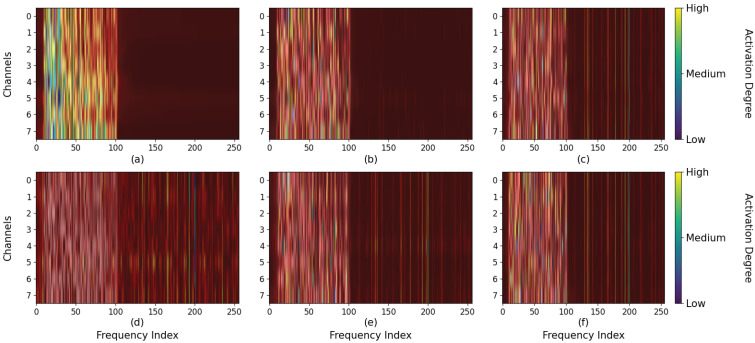
The first row of subplots (**a**–**c**) represent the class activation patterns of the model when the number of the spatial attention encoder is set to 0, 1, and 4 respectively. The second row of subplots (**d**–**f**) denotes the effective contribution region on the results of the three filtering parameters.

**Table 1 biomimetics-10-00554-t001:** Average accuracy of all methods on Benchmark dataset (%).

Method	0.5 s	1 s	1.5 s	2 s
TRCA	${29.13\pm 1.86}_{***}^{***}$	${44.54\pm 1.52}_{***}^{***}$	${52.39\pm 1.08}_{***}^{***}$	${60.32\pm 1.14}_{***}^{***}$
CCNN	${42.49\pm 1.02}_{***}^{***}$	${71.28\pm 0.66}_{***}^{***}$	${83.84\pm 0.43}_{***}^{***}$	${84.08\pm 0.39}_{***}^{***}$
SSVERNet	${43.16\pm 0.82}_{***}^{***}$	${78.35\pm 0.69}_{***}^{**}$	${89.74\pm 0.61}_{***}^{***}$	${93.54\pm 0.48}_{***}^{***}$
EEGformer	${42.92\pm 1.58}_{***}^{***}$	${75.68\pm 1.46}_{***}^{***}$	${89.54\pm 1.27}_{***}^{***}$	${92.55\pm 1.04}_{***}^{***}$
PLFA	${41.15\pm 1.40}_{***}^{***}$	${70.27\pm 1.26}_{***}^{***}$	${85.75\pm 0.88}_{***}^{***}$	${85.72\pm 0.82}_{***}^{***}$
EEGNet	${40.22\pm 1.19}_{***}^{***}$	${68.60\pm 0.94}_{***}^{***}$	${82.57\pm 0.77}_{***}^{***}$	${83.01\pm 0.68}_{***}^{***}$
SED-xLSTM	${45.83\pm 0.87}_{**}$	${80.07\pm 0.74}_{***}$	${94.62\pm 0.64}_{**}$	${96.84\pm 0.50}_{**}$
FBSED-xLSTM	47.63 ± 0.92	82.13 ± 0.68	95.97 ± 0.58	97.62 ± 0.56

**Table 2 biomimetics-10-00554-t002:** Average accuracy of all methods on BETA dataset (%).

Method	0.5 s	1 s	1.5 s	2 s
TRCA	${20.36\pm 2.40}_{***}^{***}$	${33.21\pm 2.36}_{***}^{***}$	${48.56\pm 2.14}_{***}^{***}$	${56.32\pm 1.84}_{***}^{***}$
CCNN	${32.49\pm 1.61}_{***}^{***}$	${58.55\pm 1.43}_{***}^{***}$	${71.11\pm 1.38}_{***}^{***}$	${77.63\pm 1.21}_{***}^{***}$
SSVERNet	${34.30\pm 1.43}_{***}^{*}$	${57.95\pm 0.98}_{***}^{***}$	${69.18\pm 0.85}_{***}^{***}$	${75.10\pm 0.80}_{***}^{***}$
EEGformer	${33.75\pm 1.68}_{***}^{**}$	${64.69\pm 1.34}_{***}^{***}$	${82.55\pm 1.15}_{***}^{***}$	${89.05\pm 0.82}_{***}^{***}$
PLFA	${31.18\pm 1.86}_{***}^{***}$	${57.64\pm 1.73}_{***}^{***}$	${75.07\pm 1.56}_{***}^{***}$	${79.93\pm 1.09}_{***}^{***}$
EEGNet	${25.77\pm 1.40}_{***}^{***}$	${47.24\pm 1.09}_{***}^{***}$	${58.60\pm 1.01}_{***}^{***}$	${66.84\pm 0.95}_{***}^{***}$
SED-xLSTM	${36.01\pm 1.51}_{*}$	${67.76\pm 1.08}_{**}$	${85.23\pm 0.71}_{**}$	${90.82\pm 0.77}_{**}$
FBSED-xLSTM	37.86 ± 1.04	69.31 ± 0.79	86.54 ± 0.66	92.19 ± 0.61

**Table 3 biomimetics-10-00554-t003:** Average accuracy of all methods on UCSD dataset (%).

Method	0.5 s	1 s	1.5 s	2 s
TRCA	${22.76\pm 3.32}_{***}^{***}$	${45.37\pm 3.05}_{***}^{***}$	${56.12\pm 2.89}_{***}^{***}$	${66.53\pm 2.70}_{***}^{***}$
CCNN	${39.25\pm 2.53}_{***}^{*}$	${69.18\pm 2.37}_{***}^{***}$	${80.44\pm 2.58}_{***}^{***}$	${84.17\pm 2.21}_{***}^{***}$
SSVERNet	${39.23\pm 1.57}_{***}^{**}$	${72.92\pm 1.35}_{***}^{**}$	${86.95\pm 0.94}_{***}^{*}$	${90.07\pm 0.76}_{***}^{**}$
EEGformer	${38.94\pm 1.85}_{***}^{**}$	${70.84\pm 1.50}_{***}^{***}$	${84.46\pm 1.25}_{***}^{***}$	${88.52\pm 0.97}_{***}^{***}$
PLFA	${38.02\pm 2.47}_{***}^{***}$	${68.80\pm 2.57}_{***}^{***}$	${83.95\pm 2.10}_{***}^{***}$	${87.04\pm 1.88}_{***}^{***}$
EEGNet	${37.34\pm 2.79}_{***}^{***}$	${69.71\pm 2.26}_{***}^{***}$	${83.62\pm 2.04}_{***}^{***}$	${84.24\pm 2.19}_{***}^{***}$
SED-xLSTM	${42.35\pm 1.9}_{*}$	${75.31\pm 1.59}_{*}$	${88.67\pm 1.37}_{*}$	${91.50\pm 1.03}_{**}$
FBSED-xLSTM	44.34 ± 1.73	77.57 ± 1.43	89.92 ± 1.16	93.11 ± 0.92

**Table 4 biomimetics-10-00554-t004:** For the fixed time length, the first row shows the accuracy (%) of each subset, while the second row denotes the ITR (bits/min).

Dataset	Time Length (s)	1st	2nd	3rd	4th	5th	Average
Benchmark	1 s	71.86	70.55	72.21	70.45	69.90	70.99
		122.58	118.48	123.69	118.17	116.48	119.86
	1.5 s	86.04	85.32	86.27	85.89	85.41	85.78
		114.84	112.97	115.44	114.46	113.21	114.18
BETA	1 s	60.13	59.64	60.81	60.98	59.04	60.12
		88.31	86.99	90.16	90.62	85.39	88.29
	1.5 s	77.16	76.08	76.43	75.90	76.94	76.50
		93.28	90.86	91.64	90.45	92.79	91.81
UCSD	1 s	70.26	68.95	71.73	71.19	70.47	70.52
		100.67	97.01	104.86	103.31	101.26	101.40
	1.5 s	79.87	79.32	80.91	80.06	80.46	80.12
		86.55	85.35	88.83	86.96	87.84	87.11

## Data Availability

https://github.com/dlyres/SED-xLSTM (accessed on 18 August 2025).

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
