# Peer review of "Enhanced SSVEP Bionic Spelling via xLSTM-Based Deep Learning with Spatial Attention and Filter Bank Techniques"

_biomimetics, 2025, doi:10.3390/biomimetics10080554_

Round 1
Reviewer 1 Report
Comments and Suggestions for Authors
This paper proposes a deep learning-based SSVEP speller system, termed SED-xLSTM, designed to enhance classification accuracy and information transfer rate by leveraging an extended LSTM (xLSTM) architecture, spatial attention, and filter bank techniques. The authors utilize a stacked encoder-decoder framework incorporating spatial-frequency representations through spectrogram inputs. Three public datasets are employed to benchmark the system's performance under varying time windows. The model is shown to outperform six baseline methods across classification accuracy and ITR, particularly at longer time intervals. Ablation studies and comparisons with other sequential models further support the proposed design choices. The code is publicly accessible, promoting reproducibility.
Strengths
The study addresses computation redundancy in transformer-based models by integrating xLSTM, which improves efficiency in sequence processing. The architectural design includes both spatial attention and a filter bank mechanism, which allow for improved utilization of SSVEP signal harmonics and spatial-frequency relationships. The inclusion of three publicly available datasets and a consistent benchmarking methodology enhances the credibility of results. The paper provides detailed preprocessing and implementation settings, including parameter tuning and ablation tests, ensuring transparency. Comparisons to a comprehensive suite of baselines across multiple metrics, including ITR and accuracy, further solidify the empirical grounding of the study.
Weaknesses
The rationale for selecting 16 stimuli from 40 available signals is not explained. The paper does not provide justification for this subset selection, which may bias the experimental results or limit the generalizability of findings across full-scale speller use cases.
The model fails to perform effectively in short time windows. As acknowledged in the discussion, the SED-xLSTM underperforms in the 0.5s time condition, with significant drops in both accuracy and ITR. This limitation hinders real-time applications where rapid interaction is critical.
The model ignores graph-structured relationships among EEG channels. Authors are suggested to discuss the recent graph base methods such as “Dynamic decomposition graph convolutional neural network for SSVEP-based brain–computer interface. Neural Networks” and discuss the explainability (“Trustworthy Graph Neural Networks: Aspects, Methods, and Trends. PIEEE”) of the proposed method to improve its reliability.
Reviewer 2 Report
Comments and Suggestions for Authors
The paper presents a development of a method for recognizing brain EEG patterns in the task of virtual keyboard input. The method is based on LSTM time series classification, the input is preprocessed by filters using transformer ideology of spatial attention. There are the following drawbacks.
1. It is advisable to elaborate a bit more on the problem being solved. What is the task being solved by developers (of the keyboard)? What is the task being solved by participants? What are the frequencies in Fig.1 and how do they appear? Not all of the readers are familiar with the subject area. The explanation can be done just in the beginning of the section 2.
2. The sense of processing described in lines 223-228 and Figure 4 is unclear. Why do we need splitting the signal into subbands (by frequency) the making separate FFTs for each subband then merging the spectra? The basic way to extract the subband is to make FFT, zero out unnecessary frequencies (obtaining simply one of the 'Spectrogram 1,2,3' in Figure 4), and make inverse FFT. Thus all the processing before 'Trainable Conv' in Figure 4 might be equal to simply making FFT of the source signal (and possibly zeroing out some frequencies). Please explain the need for your intricate processing and add more details in the text.
3. The developed method shows good results indeed. However it is more complex then the analogues. So good results may come not from good design but from more trainable parameters. Please give the number of trainable parameters of your method and those for analogues.
4. Continuing item 3, the suspicion for over-fitting arises. You should provide some evidence this is not the case. One of the ways is the following. The test databases contain multiple persons. Please verify that in any fold of cross-validation the data from each single person is present only in train, or in test database.
5. It is not quite clear why is the paper submitted to Biomimetics. There is no mimicking of biological objects.
Minors.
1. Line 85. 'remove' -> 'removes'.
2. It is advisable to disclose the meaning of SED abbreviation.
Reviewer 3 Report
Comments and Suggestions for Authors
This paper presents SED-xLSTM, a neural network architecture for decoding steady-state visual evoked potentials (SSVEP) in brain-computer interface (BCI) speller systems. The model combines an extended LSTM (xLSTM), spatial attention, and a filter bank within a stacked encoder-decoder framework to process spatial-frequency features and model temporal dependencies. The approach is evaluated on three public SSVEP datasets, and its performance is compared with six existing methods, including both traditional and deep learning-based models. An ablation study is conducted to assess the impact of individual model components. Several issues require revision and clarification:
-
This paper claims to be the first to apply the xLSTM to the SSVEP spelling task, but it lacks a rigorous theoretical justification for why the xLSTM is more suitable for frequency-domain EEG decoding than existing models such as the GRU or the Transformer. The paper describes architectural differences in the xLSTM (such as matrix-based gating and parallelism), but beyond general statements about long-term dependencies, it remains unclear how these differences directly benefit the EEG decoding task. A more precise analysis is needed to substantiate this architectural choice—perhaps by demonstrating how the xLSTM handles temporal dynamics, harmonics, or noise artifacts in SSVEP. Comparative visualizations or theoretical complexity comparisons would enhance its novelty.
-
The evaluation focuses primarily on within-subject performance. Despite the use of cross-validation, the paper lacks detailed results on cross-subject generalization—a crucial component for real-world BCI deployment.
-
Using FFT to construct spectrograms from EEG signals is a core component of the method, but the paper does not clarify key preprocessing steps—such as windowing strategies, overlap, and baseline correction—which can significantly impact frequency resolution and downstream model performance. Furthermore, the transformation from the time domain to the frequency domain (and its impact on spatial attention design) is conceptually incomplete.
-
Although ablation studies validate the contributions of the individual modules (SAE, EM module, and DM module), the paper does not provide metrics related to model size, inference speed, or computational efficiency, which are particularly important. I recommend conducting experiments to validate these metrics.
-
The paper emphasizes improved feature extraction through spatial attention and filter bank techniques, but does not provide qualitative evidence (such as feature heatmaps, attention maps, or class activation patterns) to demonstrate how these modules enhance discriminative power.
Round 2
Reviewer 2 Report
Comments and Suggestions for Authors
The authors have checked the issues and revised their work accordingly.
One small issue remains from previous comment #2: please explain what is the method of subband extraction.
The paper can be published after then.
Author Response
Thanks for your valuable comment. We have rewritten the third sentence of the first paragraph in Subsection 4.3 (Page 8, Line 274) to elaborate that the method of subband extraction is through bandpass filters. The revised sentence is as follows:
"To make the frequency index on the spectrogram express the feature information as richly as possible, we first filter the signal into n bands through bandpass filters, and subsequently perform FFT on the n groups of signals to obtain the frequency-domain features."
Reviewer 3 Report
Comments and Suggestions for Authors I agree to accept this paper.Author Response
The authors are gratitude for your patient reading.